# Scalable Permutation Invariant Multi-Output Gaussian Processes for Cancer Drug Response

**Leiv Rønneberg**
Department of Mathematics
University of Oslo
0851 Oslo, Norway
`ltronneb@math.uio.no`

**Vidhi Lalchand**
Eric and Wendy Schmidt Center
Broad Institute of MIT and Harvard
MIT
Cambridge, MA 02142, USA
`vidrl@mit.edu`

## Abstract

Dose-response prediction in cancer is a critical step to assessing the efficacy of drug combinations on cancer cell-lines. The efficacy of a pair of drugs can be expressively modelled through a dose-response surface which outputs the viability score across a spectrum of drug concentrations for each pair of drugs in the training data. Using large in-vitro drug sensitivity screens, the goal is to develop accurate predictive models that can be used to inform treatment decisions by predicting the efficacy of given drug combination on new cancer cell lines as well as predict the effect of unseen drugs. Previous work [15] proposed a framework for modelling dose response surfaces with multi-output GPs, however, the model relied on the exact GP marginal likelihood and prohibited scalable inference. Further, the only inputs were drug concentrations per pair while the triplet of cell-lines and drug pair corresponded to different outputs . We make two important innovations in this work, we propose a framework for stochastic multi-output GPs for scalable inference; and, use a deep generative model (DGM) to embed the drugs in a continuous chemical space - enabling viability predictions for unseen drugs. We demonstrate the performance of our model in a simple setting using a high-throughput dataset and show that the model is able to efficiently borrow information across outputs.

## 1 Introduction

In recent years, much work has gone in to developing predictive models for cancer treatments. Using data from high-throughput *in-vitro* experiments on multiple drugs and cell lines, large models are trained with the goal of predicting the sensitivity of a drug on a certain cell line. In the context of drug combinations, interest has frequently been on predicting a summary measure of drug interaction, e.g. a synergy score [11], computed from fitted dose-response surfaces, and an assumption of how non-interacting drugs should behave. Synergy scores are inherently quite crude measurements of drug interaction, and fundamentally hinge on the choice of non-interaction assumption. For this reason, several authors [7, 18, 15, 5] have proposed algorithms that instead aim to predict the entire dose-response surface. Since dose-response experiments are naturally invariant to the ordering of the drugs in a combination, we further augment the model by directly encoding this invariance into the prior.

In this paper, we build on previous work on permutation invariant multi-output Gaussian Processes (PIMOGPs) by [15]; the permutation invariance arises naturally in the context of dose-response functions as these are invariant to the ordering of the drugs in a combination. The original PIMOGPs framework relied on using exact GPs and thus incurred cubic complexity. Further, it did not encode

Workshop on Bayesian Decision-making and Uncertainty, 38th Conference on Neural Information Processing Systems (NeurIPS 2024).

the drug pairs i.e. each drug combination experiment (cell line, drug A, drug B) triplet was considered an output, and only the concentrations $(c_A, c_B)$ were given as inputs. In this work we instead take only the cell line as output, and regard the drugs as inputs alongside the drug concentrations. In order to encode the drug information as inputs we make use of a deep generative model that takes as input a string representation of the molecule, and outputs a low-dimensional representation of the drug (see A).

The main contribution of this work is deriving the stochastic variational bound for the permutation invariant linear model of coregionalisation (LMC). This method naturally handles missing data and provides uncertainty quantification.

## 2 Background

### 2.1 Linear Model of Coregionalisation for MOGPs

Multi-output GPs (MOGPs) are the extension of GP regression to the setting where the regression outputs are multidimensional, i.e. for any input $\boldsymbol{x}$, the resulting mapping $f(\boldsymbol{x}) \in \mathbb{R}^m$ for some $m > 1$. There are many ways of constructing MOGPs (see [2] for a review), but our focus here will be on the linear model of coregionalisation (LMC).

In LMC the outputs are modelled as linear combinations of a set of independent latent functions, that are themselves modelled as GPs. That is, considering a set of $m$ outputs $\{f_j(\boldsymbol{x})\}_{j=1}^m$ for an input $\boldsymbol{x} \in \mathbb{R}^p$, the $j$-th output is modelled as,

$$f_j(\boldsymbol{x}) = a_{j1}u_1(\boldsymbol{x}) + a_{j2}u_2(\boldsymbol{x}) + \cdots + a_{jR}u_R(\boldsymbol{x}), \tag{1}$$

where $u_r \sim \mathcal{GP}(0, k_r(\cdot, \cdot))$ for $r = 1, \ldots, R$ independently, and $a_{j1}, \ldots, a_{jR}$ are scalar weights. Note that each latent function is given its own covariance function $k_r$, but these are free to have the same covariance, while maintaining independence. Latent functions that share their covariance functions can be grouped into $G$ groups with $R_g$ latent functions in each group, and the equation rewritten as:

$$f_j(\boldsymbol{x}) = \sum_{g=1}^{G} \sum_{r=1}^{R_g} a_{jg}^{(r)} u_g^{(r)}(\boldsymbol{x}).$$

Since GPs are closed under addition, this construction induces a GP over all outputs. The cross-covariance between two evaluations $f_j(\boldsymbol{x})$ and $f_{j'}(\boldsymbol{x}')$ can be written as

$$\mathrm{Cov}\left[f_j(\boldsymbol{x}), f_{j'}(\boldsymbol{x}')\right] = \sum_{g=1}^{G} \sum_{r=1}^{R_g} a_{jg}^{(r)} a_{j'g}^{(r)} k_g(\boldsymbol{x}, \boldsymbol{x}') = \sum_{g=1}^{G} b_{jj'}^{(g)} k_g(\boldsymbol{x}, \boldsymbol{x}'),$$

where $b_{jj'}^{(g)} = \sum_{r=1}^{R_g} a_{jg}^{(r)} a_{j'g}^{(r)}$. The full cross-covariance over all $m$ outputs can be written as

$$\mathbf{K}(\boldsymbol{x}, \boldsymbol{x}') = \sum_{g=1}^{G} B_g k_g(\boldsymbol{x}, \boldsymbol{x}'), \tag{2}$$

where the matrix $B_g$ is known as a *coregionalisation* matrix, with entries $\{B_g\}_{ij} = b_{ij}^{(g)}$. In the case of a *complete* dataset where every input is observed at every output, the above expression can be written using Kronecker products:

$$\mathbf{K}(X, X) = \sum_{g=1}^{G} B_g \otimes K_g, \tag{3}$$

where $K_g$ has entries $\{K_g\}_{ij} = k_g(\boldsymbol{x}_i, \boldsymbol{x}_j)$. In the simplest setting, where all latent functions share the same covariance function (i.e. $G = 1$), this model is known as the *intrinsic coregionalisation model* (ICM) and the Kronecker structure can be exploited to obtain large computational gains.

For the LMC, the matrices are even larger due to the covariance across outputs, resulting in a cubic complexity in both inputs and outputs – $\mathcal{O}(n^3 m^3)$. In the special case of the IMC, i.e. where $G = 1$ in equation (3), using various Kronecker tricks can bring this down to $\mathcal{O}(n^3 + m^3)$ – see e.g. [16].

# 3 Stochastic Variational Linear model of Coregionalisation for Multi-output GPs

In this section we desribe our main insight of leveraging SVI in the setting of the LMC in order to speed up the necessary computations.

Instead of having a single set of inducing inputs, we allow different inducing locations per latent function $\mathbf{Z} = \{Z_r\}_{r=1}^R$, where $Z_r = \{\mathbf{z}_{1r}, \ldots, \mathbf{z}_{qr}\}$, with corresponding inducing variables for each latent function $\mathbf{U} = [\mathbf{u}_1^T, \ldots, \mathbf{u}_R^T]^T$, where $\mathbf{u}_r = [u_r(\mathbf{z}_{1r}), \ldots, u_r(\mathbf{z}_{qr})]^T$. In order to not overload the notation, we assume the same number of inducing variables, $q$, per latent function, and that the dataset is complete, i.e. that every input $\boldsymbol{x}$ is observed at every output. We denote by $\mathbf{Y}$ and $\mathbf{F}$ vectors of observations and latent evaluations, stacked for each output, i.e. $\mathbf{Y} = [\mathbf{y}_1^T, \ldots, \mathbf{y}_m]^T$ where $\mathbf{y}_j = [y_{1j}, \ldots, y_{nj}]^T$, where $y_{ij} = f_j(\boldsymbol{x}_i) + \epsilon_{ij}$, and similarly for $\mathbf{F}$.

The variational lower bound in the LMC case takes the same form as the stochastic variational bound for single output regression (see A.1), factorising over observations and outputs, but the matrices involved are more structured:

$$\log p(\mathbf{y}|\boldsymbol{\theta}) \geq \sum_{i=1}^n \sum_{j=1}^m \left\{ \log \mathcal{N}(y_{ij}|\boldsymbol{\alpha}_{ij}^T\tilde{\boldsymbol{\mu}}, \sigma^2) - \frac{1}{2\sigma^2}\boldsymbol{\alpha}_{ij}^T\tilde{\Sigma}\boldsymbol{\alpha}_{ij} - \frac{1}{2\sigma^2}\tilde{Q}_{ij} \right\} - \mathrm{KL}(p(\mathbf{U})\|q(\mathbf{U})), \quad (4)$$

where $\boldsymbol{\alpha}_{ij}^T = A_j K_{\mathrm{fu}}^{(i)} K_{\mathrm{uu}}^{-1}$, $A$ is the matrix with LMC coefficients, i.e. $\{A\}_{ij} = a_{ij}$, where $A_j$ denotes the $j$-th row of $A$. The matrix $K_{\mathrm{fu}}^{(i)}$ is an $R \times Rq$ block-diagonal matrix with entries $\{k_r(\boldsymbol{x}_i, Z_r)\}_{r=1}^R$, i.e. formed by evaluating the input point $\boldsymbol{x}_i$ against all the inducing points $Z_r$ in the $r$-th latent function. $\tilde{Q}_{ij}$ is formed by $A_j[K_{\mathrm{ff}}^{(i)} - K_{\mathrm{fu}}^{(i)} K_{\mathrm{uu}}^{-1} K_{\mathrm{uf}}^{(i)}]$, where the matrix $K_{\mathrm{ff}}^{(i)}$ is an $R \times R$ block-diagonal matrix with entries $\{k_r(\boldsymbol{x}_i, \boldsymbol{x}_i)\}_{r=1}^R$, $K_{\mathrm{uf}}^{(i)} = (K_{\mathrm{fu}}^{(i)})^T$ and $K_{\mathrm{uu}}$ is an $Rq \times Rq$ block-diagonal matrix with entries $\{k_r(Z_r, Z_r)\}_{r=1}^R$. Finally, $\tilde{\boldsymbol{\mu}}$ and $\tilde{\Sigma}$ come from the variational distribution $q(\mathbf{U})$ – which we assume factorise over the $r$ latent functions $q(\mathbf{U}) = \prod_{r=1}^R q_r(\mathbf{u}_r)$ where $q_r(\mathbf{u}_r) = \mathcal{N}(\boldsymbol{\mu}_r, \Sigma_r)$, yielding a block-diagonal $\tilde{\Sigma}$, and a decomposition of the KL-divergence term as $\mathrm{KL}(p(\mathbf{U})\|q(\mathbf{U})) = \sum_{r=1}^R \mathrm{KL}(p_r(\mathbf{u}_r)\|q_r(\mathbf{u}_r))$.

## 3.1 Permutation Invariance

For the problem of dose-response prediction, interest is on encoding an invariance on the ordering of the drugs — a *permutation invariance* of the inputs. In the context of multi-output GPs, we want to encode a permutation invariance for every output, $f_j$. Looking at the construction of the LMC in equation (1), it suffices to ensure that each latent function $u_r$ has the required invariance. Letting $\tilde{\boldsymbol{x}}$ denote a permuted version of the input $\boldsymbol{x}$, with the desired invariance to encode $u_r(\boldsymbol{x}) = u_r(\tilde{\boldsymbol{x}})$, this can be achieved by introducing another function $\tilde{u}_r(\boldsymbol{x})$ and constructing $u_r(\boldsymbol{x})$ via a summation argument:

$$u_r(\boldsymbol{x}) = \tilde{u}_r(\boldsymbol{x}) + \tilde{u}_r(\tilde{\boldsymbol{x}}), \quad (5)$$

from which we see that the mapping $\boldsymbol{x} \to \tilde{\boldsymbol{x}}$ leaves the function unchanged. Placing a zero-mean GP prior on $\tilde{u}_r$ with kernel $\tilde{k}_r(\cdot, \cdot)$ induces a zero-mean GP prior on $u_r$ with kernel,

$$k_r(\boldsymbol{x}, \boldsymbol{x}') = \tilde{k}_r(\boldsymbol{x}, \boldsymbol{x}') + \tilde{k}_r(\boldsymbol{x}, \tilde{\boldsymbol{x}}') + \tilde{k}_r(\tilde{\boldsymbol{x}}, \boldsymbol{x}') + \tilde{k}_r(\tilde{\boldsymbol{x}}, \tilde{\boldsymbol{x}}'). \quad (6)$$

In order to enable SVI in the context of permutation invariant MOGPs, a slight modification needs to be made regarding the inducing variables $\mathbf{u}_r$. Specifically, instead of regarding $\mathbf{u}_r$ as observations from the latent function $u_r$, they are assumed as observations from the underlying $\tilde{u}_r$. The structure of the bound in equation (4) remains unchanged. Only, the entries of matrices $K_{\mathrm{ff}}^{(i)}$, $K_{\mathrm{fu}}^{(i)}$ and $K_{\mathrm{uu}}$ need to be computed according to the following equations:

$$\begin{aligned} K_{\mathrm{ff}}^{(i)} &: k_{r,\mathrm{ff}}(\boldsymbol{x}_i, \boldsymbol{x}_i) = \tilde{k}_r(\boldsymbol{x}_i, \boldsymbol{x}_i) + \tilde{k}_r(\boldsymbol{x}_i, \tilde{\boldsymbol{x}}_i) + \tilde{k}_r(\tilde{\boldsymbol{x}}_i, \boldsymbol{x}_i) + \tilde{k}_r(\tilde{\boldsymbol{x}}_i, \tilde{\boldsymbol{x}}_i) \\ K_{\mathrm{fu}}^{(i)} &: k_{r,\mathrm{fu}}(\boldsymbol{x}_i, Z_r) = \tilde{k}_r(\boldsymbol{x}_i, Z_r) + \tilde{k}_r(\tilde{\boldsymbol{x}}_i, Z_r) \\ K_{\mathrm{uu}} &: k_{r,\mathrm{uu}}(Z_r, Z_r) = \tilde{k}_r(Z_r, Z_r). \end{aligned} \quad (7)$$

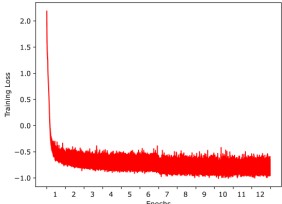 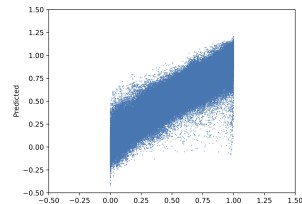 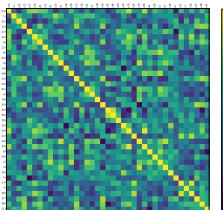

Figure 1: (Left): Training loss over 12 epochs, (Middle): Observed vs. Predicted using the permutation-invariant model, (Right): Coregionalisation matrix $B$ normalised as a correlation matrix showing the learned correlation structure across the 39 cell lines.

That is, the entries of these block diagonal matrices are computed according to updated covariance functions $k_{r,\text{ff}}(\cdot,\cdot)$, $k_{r,\text{fu}}(\cdot,\cdot)$ and $k_{r,\text{uu}}(\cdot,\cdot)$ that are themselves functions of different evaluations of the underlying kernel function $\tilde{k}_r(\cdot,\cdot)$ of $\tilde{u}_r$. This model is implemented within the GPyTorch [3] framework for GP regression.

# 4 Dataset and results

## 4.1 Dose-response data

We use the data from [12], and follow the same pre-processing procedure as in [15]. This dataset consist of 583 unique combinations of 38 drugs, screened on 39 cell lines across 6 different tissues – totalling over 1.2 million viability measurements. The pre-processing procedure (see A.3) further standardises and upsamples the data to a common $10 \times 10$ grid of concentrations — individually scaled to the unit box $[0,1] \times [0,1]$ – yielding a dataset of 1,883,700 observations on a shared grid of concentrations. The viability measurements are also standardised to the $[0,1]$ interval.

## 4.2 Results

We test the performance of our model in the *leave-triplet-out* (LTO) setting, using the nomenclature of [1]. That is, we consider prediction of a specific (cell line, drug A, drug B)-triplet that does not appear in the training dataset – however, the training dataset may contain other examples using the same cell line, or the same drugs. We split the 18,837 experiments 80/20 into a training and test set – keeping 15,069 experiments for training and 3768 for testing. For 39 cell line outputs, we set the number of latent functions $R = 10$ in the LMC all sharing the same kernel function (i.e. G=1 in equation (3)), and use $q = 200$ inducing points for each latent function. For the variational distribution, each component is modelled using a mean-field approximation, $p_r(\boldsymbol{\mu}_r, \Sigma_r)$, where $\Sigma_r$ is a diagonal matrix. For the covariance function $\tilde{k}(\boldsymbol{x}, \boldsymbol{x}')$ we use an RBF kernel over the drug concentrations, and a RBF kernel with automatic relevance determination (ARD) over the drug features. These are then multiplied together to form the final covariance function over the inputs.

We used a batch size of 256, and trained for 12 epochs using the Adam optimizer [8] with an initial learning rate of 1e-2 decreasing to 1e-3 after 6 epochs, and to 1e-4 after 9 epochs. We plot the training loss over the epochs in Figure 1 (Left). As performance metrics, we compute the root mean squared error (RMSE) and Pearson's correlation coefficient of our predictions against the observed viability measurements.

The results are visualised in Figure 1 (middle), where the model predictions are plotted against observed in the LTO setting. The model performs well, obtaining a RMSE of 0.1015 and a Pearson's r of 0.945. The pre-processing leaves us with observations strictly in the interval $[0,1]$, while the predictions have not been bounded in any way. This could be alleviated by e.g. considering a different likelihood function. Finally, we also plot the coregionalisation matrix $B$ from equation 3 having normalised it to a correlation matrix. We note that in contrast to the model of [15], our implementation is able to borrow strength across cell lines in its predictions.

## Acknowledgments and Disclosure of Funding

L.R acknowledges funding from the European Union's Horizon 2020 Research and Innovation Programme under Grant Agreement No. 847912 (RESCUER), and under the Marie Skłodowska-Curie grant agreement No. 101126636 (DSTrain). V.L acknowledges funding from the Eric and Wendy Schmidt Center postdoctoral fellowship.

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

# A   Appendix

## A.1   Stochastic Variational Gaussian Process regression

Gaussian Processes (GPs) provide a flexible non-parametric [13] paradigm for probabilistic regression and classification. They serve as versatile priors over functions fully characterised by their mean $m(\cdot)$ and covariance functions $k_\theta$; the latter typically parameterised by hyperparameters $\boldsymbol{\theta}$ which are learned from the data. When evaluated at fixed inputs we get a finite dimensional Gaussian prior over for finite values of the process $\boldsymbol{f} = (f(\boldsymbol{x}_n))_{n=1}^N$,

$$f(\boldsymbol{x}) \sim \mathcal{GP}(m(\boldsymbol{x}), k(\boldsymbol{x}, \boldsymbol{x}')) \qquad p(\boldsymbol{f}) = \mathcal{N}(\boldsymbol{m}, K_{\text{ff}}) \tag{8}$$

where $K_{\text{ff}}$ is the $N \times N$ matrix resulting from the evaluation of the kernel function $k_\theta$ on all pairs of inputs in the training dataset and similarly $\boldsymbol{m}$ characterises the mean vector resulting from the application of the mean function $m$ on each input. In the canonical regression setting we are given observations, typically a dataset $\mathcal{D}$ in the form of input-output pairs $(X, \boldsymbol{y}) = \{\boldsymbol{x}_n, y_n\}_{n=1}^N$ where $y_n$ are noisy realizations of some latent function values corrupted with Gaussian noise, $y_n = f(\boldsymbol{x}_n) + \epsilon_n$, $\epsilon_n \sim \mathcal{N}(0, \sigma^2)$. The vector $\boldsymbol{\theta}$ includes both kernel hyperparameters and a scalar noise variance $\sigma^2$. Exact inference entails maximisation of the GP marginal likelihood to obtain point estimates for $\boldsymbol{\theta}$. A Gaussian noise setting facilitates an analytically tractable marginal likelihood,

$$p(\boldsymbol{y}|\boldsymbol{\theta}) = \int p(\boldsymbol{y}|\boldsymbol{f}, \boldsymbol{\theta})p(\boldsymbol{f}|\boldsymbol{\theta})d\boldsymbol{f} = \int \mathcal{N}(\boldsymbol{y}|\boldsymbol{f}, \boldsymbol{\theta})\mathcal{N}(\boldsymbol{f}|\boldsymbol{0}, K_{\text{ff}})d\boldsymbol{f} = \mathcal{N}(\boldsymbol{y}|\boldsymbol{0}, K_{\text{ff}} + \sigma^2 I), \tag{9}$$

however, computing the exact marginal likelihood $p(\boldsymbol{y}|\boldsymbol{\theta})$ is prohibitive for larger $N$ as it incurs a $\mathcal{O}(N^3)$ computational cost dominated by the need to invert the Gram matrix $K_{\text{ff}}$. The seminal work of [17] and later [4] proposed the inducing variable framework to side-step this cubic scaling. Inducing variables $\boldsymbol{u} = \{f(\boldsymbol{z}_m)\}_{m=1}^M \subseteq \mathbb{R}$ contain values of the function at inducing inputs $Z = \{\boldsymbol{z}_m\}_{m=1}^M, \boldsymbol{z}_m \in \mathbb{R}^d$, critically $M \ll N$. Further, the posterior over function values $p(\boldsymbol{f}|\boldsymbol{y})$ is not closed form and requires variational treatment. The joint probability model in the inducing variable framework is given as,

$$p(\boldsymbol{y}, \boldsymbol{f}, \boldsymbol{u}|\boldsymbol{\theta}) = p(\boldsymbol{y}|\boldsymbol{f}, \boldsymbol{\theta})p(\boldsymbol{f}|\boldsymbol{u}, \boldsymbol{\theta})p(\boldsymbol{u}|\boldsymbol{\theta}). \tag{10}$$

and both frameworks [17, 4] use variational inference to approximate the true posterior over unknowns, $p(\boldsymbol{f}, \boldsymbol{u}|\boldsymbol{y}) \approx q(\boldsymbol{f}, \boldsymbol{u}) = p(\boldsymbol{f}|\boldsymbol{u})q(\boldsymbol{u})$. The generative model following standard Gaussian process identities consists of the likelihood $p(\boldsymbol{y}|\boldsymbol{f}) = \prod_{n=1}^N \mathcal{N}(y_n|f_n, \sigma^2)$, the conditional prior over the latent observations $p(\boldsymbol{f}|\boldsymbol{u}) = \mathcal{N}(\boldsymbol{f}|K_{\text{fu}}K_{uu}^{-1}\boldsymbol{u}, K_{\text{ff}} - K_{\text{fu}}K_{\text{uu}}^{-1}K_{\text{uf}})$, and the prior over the inducing variables $p(\boldsymbol{u}) = \mathcal{N}(\boldsymbol{u}|\boldsymbol{0}, K_{\text{uu}})$.

In the [4] scheme parameters of $q(\boldsymbol{u}) = \mathcal{N}(\boldsymbol{u}|\boldsymbol{m}, S)$ are chosen to minimise the KL divergence between the variational approximation and thr true posterior; this is tantamount to maximising the evidence lower bound (ELBO):

$$\log p(\boldsymbol{y}|\boldsymbol{\theta}) \geq = \sum_{n=1}^N \left\{ \log \mathcal{N}(y_n|\boldsymbol{k}_n^T K_{\text{uu}}^{-1}\boldsymbol{m}, \sigma^2) - \frac{1}{2\sigma^2}\text{Tr}(SK_{\text{uu}}^{-1}\boldsymbol{k}_n\boldsymbol{k}_n^T K_{\text{uu}}^{-1}) - \frac{1}{2\sigma^2}\text{Tr}(\tilde{k}_{nn}) \right\}, \tag{11}$$

where $\boldsymbol{k}_n^T$ is the $n^{th}$ row of $K_{\text{fu}}$ and $\tilde{k}_{nn}$ is the $n^{th}$ element on the diagonal of the matrix $K_{\text{ff}} - K_{\text{fu}}K_{\text{uu}}^{-1}K_{\text{uf}}$; both these terms are only dependent on their respective data-point $\boldsymbol{x}_n$. One of the main advantages of the this bound is that it can be optimized in a stochastic fashion by taking mini-batches of data making them applicable to large scale datasets.

### A.1.1 Derivation of the SVGP bound

Simplifying the KL between the variational and true posterior - $\text{KL}(q(\boldsymbol{f}, \boldsymbol{u})||p(\boldsymbol{f}, \boldsymbol{u}|\boldsymbol{y}))$ yields the following lower bound.

$$\log p(\boldsymbol{y}|\boldsymbol{\theta}) \geq \underbrace{\mathbb{E}_{q(\boldsymbol{f},\boldsymbol{u})}[\log p(\boldsymbol{y}|\boldsymbol{f})]}_{\mathcal{L}_1} - \text{KL}(q(\boldsymbol{u}|\boldsymbol{\theta})||p(\boldsymbol{u}|\boldsymbol{\theta})). \tag{12}$$

If the first term above entails a factorisable likelihood, then the expectation w.r.t the variational distribution $q(\boldsymbol{f}, \boldsymbol{u}) = p(\boldsymbol{f}|\boldsymbol{u}, \boldsymbol{\theta})q(\boldsymbol{u}|\boldsymbol{\theta})$ is analytically tractable yielding,

$$
\begin{aligned}
\mathcal{L}_1 &= \int q(\boldsymbol{u}|\boldsymbol{\theta}) \int p(\boldsymbol{f}|\boldsymbol{u}, \boldsymbol{\theta}) \log \prod_{n=1}^{N} p(y_n|f_n) d\boldsymbol{f} d\boldsymbol{u} \\
&= \int q(\boldsymbol{u}|\boldsymbol{\theta})[\log \mathcal{N}(\boldsymbol{y}|K_{\text{fu}}K_{\text{uu}}^{-1}\boldsymbol{u}, \sigma^2 I) - \frac{1}{2\sigma^2}\text{Tr}(K_{\text{ff}} - K_{\text{fu}}K_{\text{uu}}^{-1}K_{\text{uf}})]d\boldsymbol{u} \\
&= \sum_{n=1}^{N} \left\{ \log \mathcal{N}(y_n|\boldsymbol{k}_n^T K_{\text{uu}}^{-1}\boldsymbol{m}, \sigma^2) - \frac{1}{2\sigma^2}\text{Tr}(SK_{\text{uu}}^{-1}\boldsymbol{k}_n\boldsymbol{k}_n^T K_{\text{uu}}^{-1}) - \frac{1}{2\sigma^2}\text{Tr}(\tilde{k}_{nn}) \right\},
\end{aligned}
\tag{13}
$$

where $\boldsymbol{k}_n^T$ is the $n^{th}$ row of $K_{\text{fu}}$ and $\tilde{k}_{nn}$ is the $n^{th}$ element on the diagonal of the matrix $K_{\text{ff}} - K_{\text{fu}}K_{\text{uu}}^{-1}K_{\text{uf}}$; both these terms are only dependent on their respective data-point $\boldsymbol{x}_n$.

## A.2 Continuous representation of drugs

In the PIICM, each drug combination experiment (cell line, drug A, drug B) triplet was considered an output, and only the concentrations $(c_A, c_B)$ was given as inputs. In this manuscript we instead take only the cell line as output, and regard the drugs as inputs alongside the drug concentrations. In order to encode the drug information as inputs, we could one-hot encode them, but instead make use of a deep generative model that takes as input a string representation of the molecule, and outputs a low-dimensional representation of the drug.

### A.2.1 SMILES v. SELFIES

Most of the literature on machine learning based chemical design for the string representation of molecules uses SMILES strings [19] — a line notation method which encodes molecular structure using short ASCII strings. However, the SMILES representation has two critical limitations. First, they are not designed to capture molecular similarity, hence molecules with almost identical structure can have markedly different SMILES [6]. Second, they are not robust on their own, which means that generative models are likely to produce strings that do not represent valid molecules. Hence, the latent space of DGMs trained on SMILES strings can potentially have large dead zones where none of the points sampled in the region decode to valid molecules. To overcome these issues, we train our model on an alternative string representation for molecules introduced in 2020 [10] that guarantees that guarantees 100% robustness — SELF-referencing embedded string (SELFIES). We do not deep dive into technical construction aspects of the SELFIE syntax in this work, at a high-level one of the difficulties of working with SMILES is the nested bracket closures which appear frequently in the SMILES notation, for instance, consider the smiles string `CCCc1cc(NC(=O)CN2C(=O)NC3(CCC(C)CC3)C2=O)n(C)n1`, the SELFIE translation uses a formal Chomsky type-2 grammar or a context-free grammar and gets rid of the non-local characteristics. The molecule above is translated to `[C][C][C][C][C][=C][Branch2][Ring1][=C][N][C]` `[=Branch1][C][=O][C][N][C][=Branch1][C][=O][N][C][Branch1][O][C][C][C]` `[Branch1][C][C][C][C][Ring1][#Branch1][C][Ring1][N][=O][N][Branch1][C][C][N]` `[=Ring2][Ring1][#Branch1]`. We tokenize the SELFIE syntax to represent molecules in our generative model.

The deep generative model used in this context is trained for autoencoding and features an open-ended chemical latent space learnt by embedding discrete molecules in a continuous vector space (encoder). For generation, an inverse step (decoder) converts a continuous vector in latent space to a valid molecule. This is the classical encoder-decoder set-up as in a standard VAE [9]. We use a recurrent VAE architecture with an RNN encoder and a decoder to sequentially process the SELFIE representation of the drugs token by token. We embed the cancer drugs in the latent space of the generative model by representing them as SELFIEs and encoding them using the trained encoder. This yields a 50-dimensional latent vector for each drug, i.e. $x_A \in \mathbb{R}^{50}$ for drug A, for example. Combined with the concentrations, the vector of inputs for each data point becomes $x = (c_A, c_B, x_A^T, x_B^T) \in \mathbb{R}^{102}$, with the corresponding permuted version $\tilde{x} = (c_A, c_B, x_B^T, x_A^T)$ that the model is invariant to. Due to the string representation of 4 cancer drugs being too long for the generative model only 34 of the 38 drugs were successfully given coordinates in the latent space. Removing experiments for which $x_A$ or $x_B$ is missing leaves 483 unique combinations, and a total of 18,837 unique experiments.

## A.3 Data pre-processing

Each experiment is processed using the bayesynergy [14] software, which fits a semi-parametrics dose-response function to the data, and provides samples from the posterior predictive dose-response function on a $10 \times 10$ grid of concentrations — individually scaled to the unit box $[0, 1] \times [0, 1]$ – yielding a dataset of 2,273,700 observations on a shared grid of concentrations. As a notable difference from the pre-processing procedure in [15], is that instead of training our model on targets derived from the latent GP in the bayesynergy software, we instead take as our targets the fitted values of the dose-response function, which takes values between 0 and 1. Removing combinations containing drugs that we could not embed in our latent space the final dataset consists of 1,883,700 unique viability measurements stemming from 18,837 unique (cell line, drug A, drug B) triplets.

