# OpenReview forum: "Scalable Permutation Invariant Multi-Output Gaussian Processes for Cancer Drug Response"
_NeurIPS.cc/2024/Workshop/BDU — NeurIPS BDU Workshop 2024 Poster_

### Official Review · Reviewer_sSQE · 2024-09-16
**an interesting paper with detailed formulation of the model. while lacks some details in experiment results**

**Rating:** 6
**Confidence:** 5

**Review:**

The paper presents a technically sound and innovative approach to dose-response prediction in cancer drug combinations using permutation-invariant multi-output Gaussian Processes (MOGPs). It effectively leverages a combination of Stochastic Variational Inference (SVI) and Deep Generative Models (DGMs) to overcome key limitations in previous works. The methodology is mathematically rigorous, employing well-established Gaussian process frameworks like Linear Model of Coregionalization (LMC) while extending them to more scalable and flexible models.
The originality of this work is high, as it introduces new solutions to known problems in cancer drug-response prediction:
	•	Permutation Invariance: Encoding permutation invariance directly into the Gaussian process framework is a notable innovation that respects the inherent symmetry in dose-response experiments.
	•	Scalability: By leveraging Stochastic Variational Inference (SVI), the paper improves scalability, addressing the prohibitive cubic complexity of traditional GP models.
	•	Use of SELFIES: Incorporating the SELFIES molecular representation to improve the robustness of chemical embeddings in drug-response models is another innovative contribution.

The only part which is not satisfying is the relatively short result sections. It could benefit from more detailed analysis and, crucially, comparisons with other SOTA methods. What is the performance compared with others? If it is not as good as others, whct is the benefits and potential strength of the proposed method?

---

### Official Review · Reviewer_tywh · 2024-09-23
**Sound approach to scalable MOGPs**

**Rating:** 7
**Confidence:** 1

**Review:**

This paper presents a novel and technically sound approach to scalable MOGPs for cancer drug response prediction, incorporating permutation invariance and deep generative models. While the results are promising, the complexity of the method and limited generalizability are difficult for me to understand.

Pros:
1. The paper highlights a new efficient method by using stochastic variation inference.
2. The results are well presented and the paper is well written.
3. The paper highlights an interesting use of deep generative models to embed drugs as part of a vector space.

Cons:
1. Difficult to understand the intricacies of the generative model.
2. Hard to understand how it will scale and what obstacles lie in practical use cases.

---

### Decision · Program_Chairs · 2024-10-09

Accept (Poster)